# Prevalence of Restless Legs Syndrome and its Symptoms among Patients with Spinal Disorders

**DOI:** 10.3390/jcm10215001

**Published:** 2021-10-27

**Authors:** Hidetomi Terai, Hiromitsu Toyoda, Masatoshi Hoshino, Akinobu Suzuki, Shinji Takahashi, Koji Tamai, Shoichiro Ohyama, Akito Yabu, Hiroaki Nakamura

**Affiliations:** 1Department of Orthopaedic Surgery, Osaka City University Graduate School of Medicine, Osaka 545-8585, Japan; hterai@med.osaka-cu.ac.jp (H.T.); a-suzuki@msic.med.osaka-cu.ac.jp (A.S.); shinji@med.osaka-cu.ac.jp (S.T.); koji.tamai.707@gmail.com (K.T.); yabuakito@gmail.com (A.Y.); hnakamura@med.osaka-cu.ac.jp (H.N.); 2Department of Orthopaedic Surgery, Osaka City General Hospital, Osaka 534-0021, Japan; hoshino717@gmail.com; 3Department of Orthopaedic Surgery, Nishinomiya Watanabe Hospital, Murokawacho 662-0863, Japan; a03ma012@yahoo.co.jp

**Keywords:** restless legs syndrome, spinal disorder, lumbar spinal stenosis, leg cramp

## Abstract

Restless legs syndrome (RLS) is a neurological disorder that causes uncomfortable sensations in the legs. The purpose of this study was to evaluate the symptoms of RLS in patients with spinal disorders and the impact of RLS on the clinical outcomes of lumbar spinal stenosis (LSS). The records of 278 patients (age range 65–92 years) with spinal disorders who visited our outpatient clinic were reviewed. We used a survey to identify subjects with RLS based on the International RLS Study Group diagnostic criteria. We further recorded patient characteristics, surgical outcomes, sleeping time, mental health condition, and the occurrence of leg cramps. Thirty-two patients (11.5%) met the criteria for RLS. The prevalence of anxiety (46.9% vs. 26.6%, *p* = 0.023) and leg cramps (90.6% vs. 73.2%, *p* = 0.030) was higher in patients with RLS than in those without. RLS was present in 12.3% of LSS patients. The visual analog scale score for lower back pain before surgery and at the final follow-up was significantly higher in LSS patients with RLS than in those without. However, the Japanese Orthopaedic Association (JOA) score, JOA score improvement ratio, and VAS score for leg numbness were not significantly different between the groups.

## 1. Introduction

Paresthesia, such as burning or prickling sensations, is a common symptom of patients with neurological problems. The incidences of lumbar spinal stenosis (LSS) and cervical spondylotic myelopathy have increased in recent years, reflecting our aging society. Consequently, patients with leg discomfort may present to spinal clinics to identify an underlying spine problem. Surgery may be recommended for patients with severe spinal stenosis to create more space for the spinal cord or nerves. However, residual symptoms, such as sensory disturbances of the extremities or leg cramps, may occur after surgery. While most patients are satisfied after LSS surgery, studies have shown that up to one-third of patients may be dissatisfied [1,2,3]. The increasing incidence of age-related changes on magnetic resonance imaging among asymptomatic aging subjects often leads to a wrong diagnosis. Indeed, an incorrect initial diagnosis may explain the prevalence of residual symptoms postoperatively. We hypothesized that restless legs syndrome (RLS) might be a cause of wrong diagnoses in patients with spinal disorders.

RLS, also known as Willis–Ekbom disease, is a common condition of the nervous system characterized by an irresistible urge to move the legs, often associated with an unpleasant sensation [4]. Although most cases of RLS are primary, RLS is often linked to other disorders, such as chronic kidney disease [5], iron deficiency [6], movement disorders including Parkinson’s disease [7], peripheral neuropathy [8], common chronic diseases [9] including arterial hypertension and headache, as well as conditions such as pregnancy [10] and inflammation [11]. RLS is frequently associated with painful polyneuropathy, but despite the typical symptoms reported by patients, it remains poorly recognized by neurologists. Many patients with RLS may be referred to spinal clinics, but only a few clinical studies have investigated this aspect so far [12,13,14]. Spine clinic physicians should be aware of conditions that are associated with secondary RLS or that mimic RLS.

Therefore, we conducted a questionnaire survey to identify subjects with RLS using the diagnostic criteria of the International RLS Study Group (IRLSSG) [15] and evaluated the prevalence of RLS and its symptoms among patients with spinal disorders and the impact of RLS symptoms on the clinical outcomes after LSS surgery. We also investigated the other clinical observations that could be associated with RLS such as sleeping time, mental health condition, and the occurrence of leg cramps.

## 2. Materials and Methods

### 2.1. Study Population

From August 2015 to July 2016, we continuously recruited spine clinic outpatients who agreed to screening for sarcopenia and RLS. The inclusion criteria were age >65 years, diagnosis of a spinal disorder, and the ability to ambulate. Patients with implanted metal or electrical devices were excluded since these might affect bioelectrical impedance analysis of muscle mass. We have previously described the relationship between sarcopenia, back muscle strength, spinopelvic parameters, and clinical outcomes [16,17,18]. At the same time, we administered a self-reported questionnaire to each patient to evaluate not only the prevalence of RLS symptoms, but also sleeping time, the frequency of nocturnal leg cramps, and the level of anxiety and depression. We focused on these variables because RLS and nocturnal leg cramps are two common causes of nocturnal leg pain or discomfort, and psychiatric disorders are associated with sleeping disorders. The study protocol was approved by the Institutional Review Board of Osaka City University (approval no. 3170), and all participants provided written informed consent prior to their enrollment. All methods were performed in accordance with the Declaration of Helsinki and the Ethical Guidelines for Medical and Health Research Involving Human Subjects in Japan. A total of 278 outpatients at our spine clinic were enrolled in this cross-sectional observational study. To evaluate the impact of RLS symptoms on the clinical outcomes after lumbar decompression surgery, we retrospectively analyzed the data of the 130 patients who underwent lumbar decompression surgery.

### 2.2. RLS Questionnaire

We used the Japanese version of the RLS questionnaire developed by the RLS Diagnosis and Epidemiology Workshop at the National Institutes of Health (NIH) in collaboration with members of the IRLSSG, with particular consideration of the recommended questions for epidemiological studies of RLS [15,19]. This NIH/IRLSSG questionnaire consists of the following four essential clinical criteria: (1) an urge to move the legs, usually accompanied or caused by an uncomfortable sensation in the legs; (2) beginning or worsening of symptoms during periods of rest or inactivity; (3) partial or total relief of symptoms by movement; and (4) symptoms that are worse in the evening or night than during the day or that occur only in the evening or night. Only patients who met all four diagnostic criteria were deemed to have RLS [20]. We also investigated the average sleeping time per day.

### 2.3. Questionnaire on Nocturnal Leg Cramps

The frequency of nocturnal leg cramps was evaluated using previously described questionnaires [21,22]: (1) Have you recently experienced cramps in the legs?; (2) How often do you have leg cramps?; (3) At what time of the day do you have leg cramps?; (4) Where do you have leg cramps most frequently?

### 2.4. Questionnaire on Anxiety and Depression

We used the validated Japanese version of the Hospital Anxiety and Depression Scale (HADS) to assess the level of anxiety and depression [23]. The HADS consists of 14 items and the anxiety (HADS-A) and depression (HADS-D) subscales each of 7 items. Each item on the questionnaire is scored from 0 (not at all) to 3 (always). Each of the seven items is scored from 0 to 21 points, and a score ≥8 on a subscale or ≥11 in the entire HADS was considered a clinically significant condition [24]. 

### 2.5. Clinical Outcomes

The records of 130 LSS patients who underwent minimally invasive lumbar decompression surgery were retrospectively reviewed to investigate the clinical impact of RLS on the outcomes after surgery. All patients underwent bilateral decompression via a unilateral approach to decompress the central and bilateral lateral recess using a microscope or the METRx^®^ Microendoscopic Discectomy System (Medtronic, Dublin, Ireland) as previously described [25,26]. Clinical outcomes were evaluated with the Japanese Orthopaedic Association (JOA) score and visual analog scale (VAS) score for lower back pain, leg pain, and leg numbness preoperatively and at the last follow-up. The JOA score was developed by the JOA to measure outcomes for patients with lower back problems. It consists of 29 points and has been widely utilized to evaluate the functional results of many types of intervention for patients with lumbar disorders [27]. The improvement rate for the JOA scores was calculated as: (postoperative JOA score−preoperative JOA score)/(29−preoperative JOA score) × 100 (%) [28]. 

### 2.6. Statistical Analysis 

All data were presented as the mean ± standard deviation with a statistical significance level set at a *p*-value < 0.05. We used IBM SPSS Statistics for Windows, version 25.0 (IBM Corp., Armonk, NY, USA) for all statistical analyses. Continuous and categorical variables were compared using the t-test and chi-squared test, respectively, between patients with RLS (RLS group) and those without RLS (no-RLS group).

## 3. Results

### 3.1. Patient Baseline Characteristics

The baseline characteristics of the patients are shown in Table 1. The mean age of patients was 72.3 years, and 50.4% were male. Preoperative diagnoses were cervical spine disorders in 54 patients (19.4%), thoracic spine disorders in 14 patients (5.0%), and lumbar spine disorders in 210 patients (75.5%).

### 3.2. Prevalence of RLS

Of the 278 patients, 32 (11.5%) were diagnosed with RLS. Overall, 56.1% of patients had an urge to move their legs, usually accompanied or caused by an uncomfortable sensation in the legs. Furthermore, 43.2% of patients had symptoms that began or worsened during periods of rest or inactivity, and 42.1% experienced partial or total relief by movement. In 31.7% of patients, symptoms were worse in the evening or night or occurred only during these times. Two hundred and seven patients (74.5%) met one or more of the RLS criteria. The average score of the IRLSSG questionnaire was 1.7 ± 1.3.

### 3.3. Comparison of Baseline Characteristics between RLS Group and no-RLS Group

There were no significant differences in the age, gender, body mass index, diagnosis, and sleeping time between the groups. However, the RLS group showed a statistically significantly higher prevalence of anxiety (HADS-A score >7) (46.9% vs. 26.6%, *p* = 0.02) and leg cramps (90.6% vs. 73.2%, *p* = 0.03) than the no-RLS group. Leg cramps occurred more frequently in the RLS group than the no-RLS group (*p* = 0.05). There were no significant differences in the prevalence of depression (HADS-D score >7) and the area affected by leg cramps.

### 3.4. Comparison of Clinical Outcomes between Patients in RLS Group and no-RLS Group Who Underwent Lumbar Decompression Surgery

The results of the comparison are shown in Table 2. Of the 130 LSS patients who underwent decompression, 16 (12.3%) had RLS at the final follow-up. The average follow-up period was 41.2 months. The JOA score, JOA score improvement ratio, and VAS score for leg numbness were not significantly different between groups. However, the preoperative lower back pain and final follow-up VAS scores were significantly higher in RLS patients than in those without RLS (65.1 vs. 43.4, *p* = 0.01 and 33.8 vs. 19.0, *p* = 0.02, respectively). No significant differences existed in the prevalence of depression and anxiety. The frequency of leg cramps was significantly higher in patients with RLS than in those without (*p* = 0.04).

## 4. Discussion

In this cross-sectional observational study, the prevalence of RLS among patients at our spinal clinic using the IRLSSG questionnaire was 11.5%. Overall, 74.5% of the patients met one or more RLS criteria. Among LSS patients, the prevalence of RLS was 12.3%. The relation between RLS and spinal disorders is unknown to date, and the most important finding of our study is that a considerable number of patients with spinal disorders have RLS. Given that RLS symptoms are relatively common in spine patients, improved clinician awareness regarding the possibility of RLS might help reduce diagnostic errors.

RLS used to be classified into idiopathic, primary, inherited, and secondary to a variety of chronic diseases [29,30]. On the other hand, “RLS mimics” refers to conditions that need to be considered in the differential diagnosis, such as arthritis, leg cramps, peripheral neuropathy, and leg edema. Some researchers have proposed that RLS is a continuous spectrum with a major genetic component on one end and primarily environmental causes or comorbidities at the other [9]. This previous distinction into “idiopathic, primary” and “secondary” RLS has been abandoned in the new Diagnostic and Statistical Manual of Mental Disorders criteria as it seems that RLS is a comorbidity. 

The estimated prevalence of RLS among adults is between 5.0% and 14.3% [31,32,33,34]. A lower prevalence, varying from 0.6% to 3.9%, has been reported in Asian populations. For instance, the prevalence of RLS in the general Japanese population is 1.8% [33]. Of note, the prevalence of RLS in several medical conditions associated with RLS is higher than that in the general population. The prevalence of RLS was found to be 14% among patients with Parkinson’s disease [35], 24.2% among patients with chronic kidney disease [5], 25–35% among patients with iron deficiency or anemia [6], and 5.2–53.7% among peripheral neuropathy [8]. However, few studies have explored the prevalence of RLS in patients with spinal disorders. Kocabicak et al. reviewed patients with lumbar radiculopathy who were resistant to conservative treatment and found an RLS prevalence of 68.1% before surgery, which was significantly higher than that of the control group (16.4%) and decreased to 24.2% after surgery [13]. Yang et al. retrospectively reviewed the medical records of 639 patients with leg discomfort undergoing spine magnetic resonance imaging, and 5% met the criteria of RLS [12]. Our study revealed an RLS prevalence of 11.5% in patients with spinal disorders, and 74.5% of all patients met one or more RLS criteria. Our study subjects were outpatients with relatively stable conditions, and more than 75% underwent spine surgery. We speculated that the prevalence of RLS in our patients with spinal disorders was among the range of prevalence found in the normal population, while higher than that in the Japanese general population. Although the presence of peripheral neuropathy (PN) should be considered as an exclusion criterion for the diagnosis of primary RLS, recent systematic reviews showed a significantly higher frequency of RLS in the PN group than in the control group [8]. The role of nerve damage in the causality and progression of RLS remains unclear and further studies are required. 

Resting numbness in the soles of both feet can occur in patients with LSS. Such numbness and the occurrence of cauda equina symptoms are predictive of poor prognosis or residual symptoms after surgery [1,36]. In cervical spondylotic myelopathy, numbness following a glove and stocking pattern is one of the most frequent complaints, and it was reported that approximately 50% of sensory disturbances persisted after surgery [37,38]. Typically, RLS symptoms differ from neuropathic symptoms resulting from spinal disorders but may sometimes be similar. If leg numbness or other symptoms, e.g., tingling, are refractory to treatment, it is important to rule out other coexisting illnesses that may explain the symptoms. In our study, 32 patients met four IRLSSG criteria but only six were treated, for example, with dopaminergic agents. Among these six patients, five experienced an improvement in their residual symptoms after surgery on medication. The awareness of RLS can help reduce diagnostic errors and it can be treated quite easily with some medications.

We compared the clinical outcomes after lumbar decompression between LSS patients with and without RLS. The JOA score was not different between these groups. However, the VAS score for lower back pain before surgery and at the final follow-up was significantly higher in patients with RLS. It was reported that the volume of dopamine in the mesolimbic dopaminergic system is decreased in chronic pain patients [39]. We had hypothesized that the low volume of dopamine associated with chronic pain and RLS or RLS-like symptoms may affect the clinical outcomes after surgery. However, our study indicated that clinical outcomes of LSS patients with RLS were acceptable. We concluded that LSS with RLS is not a contraindication for spinal surgery, even if RLS may be one of the causes of residual back pain postoperatively.

We further found that anxiety and leg cramps were more frequent in patients with RLS than in those without. RLS patients are at an increased risk of specific anxiety, depressive, and, particularly, panic disorders [30,40]. Leg cramps are more frequent in LSS patients than the control group [21]. The diagnosis of RLS and nocturnal leg cramps in patients with spinal disorders presents a clinical challenge because of the symptom overlap and incomplete understanding of the pathophysiological mechanisms. In this situation, clinicians should focus on distinguishing the characteristics of each condition for both the diagnosis and assessment of impact to reach an appropriate treatment decision.

Our results should be interpreted within the limitations of our study. First, our study was exploratory, and the study population consisted of a small number of outpatients (*n* = 269) treated at a single center. We did not perform an a priori statistical power analysis because it would have been difficult to determine the sample size since the prevalence of RLS in patients with spinal disorders was unknown. Second, we only compared our data with previous reports because of the cross-sectional study design that did not consider data from healthy subjects. Our results do not establish a cause–effect relationship between RLS and spinal disorders. Third, there was a potential source of bias at baseline, including a confounding bias. We performed this survey using the NIH/IRLSSG criteria 1–4. The occurrence of RLS symptoms is not solely accounted for as symptoms primary to another medical condition, side effect of drugs, or a behavioral condition. Consequently, our findings need to be verified in a prospective large multicenter study. Finally, as patients did not complete the RLS questionnaire before and after surgery, it is unclear whether RLS symptoms improve with the improvement of spinal disorders.

## 5. Conclusions

To the best of our knowledge, this is the first study that investigated the prevalence of RLS in patients with spinal disorders. The prevalence of RLS in outpatients with spinal disorders was 11.5% and 12.5% in those with LSS. Thus, RLS symptoms are relatively common in spine patients, and improved clinician awareness of RLS might help reduce diagnostic errors.

## Figures and Tables

**Table 1 jcm-10-05001-t001:** Baseline characteristics of 278 outpatients at a single spine center in this study on the prevalence of restless legs syndrome in spinal disorders. *RLS*—restless legs syndrome, *IRLSSG*—the international restless legs syndrome study group, *HADS*—the hospital anxiety and depression scale, *HADS-A*—HADS anxiety, *HADS-D*—HADS depression.

Variables	All	RLS (+)	RLS (−)	*p* Value
Numbers (%)	278 (100%)	32 (11.5%)	246 (88.5%)	
Male:Female	140:138	18:14	122:124	0.579
Age (years)	72.3 (10.9)	70.6 (11.1)	72.5 (10.9)	0.350
Body mass index (kg/cm^2^)	23.3 (3.5)	24.0 (3.2)	23.2 (3.6)	0.237
Diagnosis				0.816
Cervical spinal disorders (%)	54 (19.4%)	5 (9.3%)	49 (90.7%)	
Cervical spondylotic myelopathy, *n*	47	4	43	
Ossification of the posterior longitudinal ligament, *n*	4	1	3	
Intradural extramedullary tumor, *n*	3	0	3	
Thoracic spinal disorders (%)	14 (5.0%)	2 (14.3%)	12 (85.7%)	
Osteoporotic vertebral fracture, *n*	7	0	7	
Ossification of the yellow ligament, *n*	3	1	2	
Diffuse idiopathic skeletal hyperostosis, *n*	2	0	2	
Intradural extramedullary tumor, *n*	2	1	1	
Lumbar spinal disorders (%)	210 (75.5%)	25 (11.9%)	185 (88.1%)	
Lumbar spinal stenosis, *n*	163	19	144	
Degenerative lumbar scoliosis, *n*	14	1	13	
Degenerative lumbar spondylolisthesis, *n*	10	0	10	
Osteoporotic vertebral fracture, *n*	12	1	11	
Lumbar disc herniation, *n*	8	3	5	
Intradural extramedullary tumor, *n*	3	1	2	
Surgery, yes (%)	218 (78.4%)	28 (87.5%)	190 (77.2%)	0.254
IRLSSG questionnaire, yes (%)				
(1) An urge to move the legs, usually accompanied or caused by an uncomfortable sensation in the legs	156 (56.1%)	32 (100%)	124 (50.8%)	*p* < 0.001
(2) Beginning or worsening of symptoms during periods of rest or inactivity	120 (43.2%)	32 (100%)	88 (35.8%)	*p* < 0.001
(3) Partial or total relief of symptoms by movement	117 (42.1%)	32 (100%)	85 (34.6%)	*p* < 0.001
(4) Symptoms that are worse in the evening or night than during the day, or that occur only in the evening or night	88 (31.7%)	32 (100%)	56 (22.8%)	*p* < 0.001
IRLSSG questionnaire, score, yes (%)				*p* < 0.001
0	71 (25.5%)	0 (0%)	71 (28.9%)	
1	51 (18.3%)	0 (0%)	51 (20.7%)	
2	70 (25.2%)	0 (0%)	70 (28.5%)	
3	54 (19.4%)	0 (0%)	54 (22.0%)	
4	32 (11.5%)	32 (100%)	0 (0%)	
IRLSSG questionnaire, score	1.7 (1.3)	4.0 (0)	1.43 (1.1)	*p* < 0.001
Sleeping time (hours)	6.1 (1.0)	5.9 (1.1)	6.1 (1.0)	0.268
HADS-A score	5.5 (3.9)	6.8 (4.5)	5.3 (3.8)	0.052
Anxiety (cut off >7), yes (%)	80 (28.8%)	15 (46.9%)	65 (26.6%)	0.023
Anxiety (cut off >10), yes (%)	33 (11.9%)	7 (21.9%)	26 (10.7%)	0.081
HADS-D score	7.8 (3.1)	8.4 (3.2)	7.8 (3.1)	0.311
Depression (cut off >7), yes (%)	144 (51.8%)	20 (62.5%)	124 (50.8%)	0.260
Depression (cut off >10), yes (%)	53 (19.1%)	9 (28.1%)	44 (18.0%)	0.230
Leg cramp, yes (%)	209 (75.2%)	29 (90.6%)	180 (73.2%)	0.030
Leg cramp, how often, yes (%)				0.051
More than once a day	21 (7.6%)	6 (18.8%)	15 (6.1%)	
Once a day	25 (9.0%)	4 (12.5%)	21 (8.5%)	
Once every several days	69 (24.8%)	10 (31.3%)	59 (24.0%)	
Once every several weeks	47 (16.7%)	4 (12.5%)	43 (17.5%)	
Once every several months	47 (16.9%)	5 (15.6%)	42 (17.1%)	
None	69 (24.8%)	3 (9.4%)	66 (26.8%)	
Leg cramp, when, yes (%)				0.173
During sleep	126 (45.3%)	15 (46.9%)	111 (45.1%)	
In the morning	43 (15.5%)	6 (18.8%)	37 (15.0%)	
In the afternoon	14 (5.0%)	2 (6.3%)	12 (4.9%)	
In the evening	15 (5.4%)	4 (12.5%)	11 (4.5%)	
Non-regular time	11 (4.0%)	2 (6.3%)	9 (3.7%)	
None	69 (24.8%)	3 (9.4%)	66 (26.8%)	
Leg cramp, where, yes (%)				0.105
Calf	135 (48.6%)	22 (68.8%)	113 (45.9%)	
Shin	10 (3.6%)	1 (3.1%)	9 (3.7%)	
Anterior thigh	8 (2.9%)	2 (6.3%)	6 (2.4%)	
Posterior thigh	19 (6.8%)	1 (3.1%)	18 (7.3%)	
Foot	37 (13.3%)	3 (9.4%)	34 (13.8%)	
None	69 (24.8%)	3 (9.4%)	66 (26.8%)	

RLS (+): diagnosed with restless legs syndrome. RLS (−): not diagnosed with restless legs syndrome.

**Table 2 jcm-10-05001-t002:** Comparison of baseline characteristics and clinical outcomes of patients (*n* = 130) who performed lumbar decompression surgery between those with and those without restless legs syndrome. *RLS*—restless legs syndrome, *IRLSSG*—the international restless legs syndrome study group, *HADS*—the hospital anxiety and depression scale, *HADS-A*—HADS anxiety, *HADS-D*—HADS depression.

Variables	LSS with RLS	LSS without RLS	*p* Value
Numbers (%)	16 (12.3%)	114 (87.7%)	
Male:Female	6:10	64:50	0.188
Age (years)	78.2 (5.1)	76.7 (6.5)	0.387
Body mass index (kg/cm^2^)	23.6 (4.4)	23.1 (3.2)	0.584
Follow-up period (months)	37.9 (17.4)	42.2 (41.1)	0.677
JOA score			
before surgery	11.3(4.5)	12.7 (4.2)	0.231
at the final follow-up	23.2 (3.9)	24.4 (3.6)	0.230
Improvement ratio of JOA score (%)	71.1 (20.0)	70.3 (20.5)	0.886
VAS for lower back pain			
before surgery	65.1 (25.1)	43.4 (28.3)	0.010
at the final follow-up	33.8 (24.9)	19.0 (22.5)	0.020
VAS for leg pain			
before surgery	74.7 (24.5)	59.5 (27.8)	0.062
at the final follow-up	24.6 (25.0)	20.7 (28.3)	0.614
VAS for leg numbness			
before surgery	72.0 (28.1)	59.2 (29.8)	0.159
at the final follow-up	39.7 (28.1)	29.2 (30.8)	0.210
IRLSSG questionnaire, yes (%)			
(1) An urge to move the legs, usually accompanied or caused by an uncomfortable sensation in the legs	16 (100%)	72 (63.7%)	0.001
(2) Beginning or worsening of symptoms during periods of rest or inactivity	16 (100%)	47 (41.2%)	*p* < 0.001
(3) Partial or total relief of symptoms by movement	16 (100%)	42 (36.8%)	*p* < 0.001
(4) Symptoms that are worse in the evening or night than during the day, or that occur only in the evening or night	16 (100%)	34 (29.8%)	*p* < 0.001
IRLSSG questionnaire, score, yes (%)			*p* < 0.001
0	0 (0%)	23 (20.2%)	
1	0 (0%)	20 (17.5%)	
2	0 (0%)	41 (36.0%)	
3	0 (0%)	30 (26.3%)	
4	16 (100%)	0 (0%)	
IRLSSG questionnaire, score	4 (0)	1.68 (1.1)	*p* < 0.001
Sleeping time (hours)	5.9 (1.0)	6.2 (1.0)	0.348
HADS-A score	6.2 (4.9)	4.8 (3.5)	0.154
Anxiety (cut off >7), yes (%)	6 (37.5%)	25 (21.9%)	0.210
Anxiety (cut off >10), yes (%)	4 (25.0%)	9 (7.9%)	0.056
HADS-D score	7.8 (2.5)	7.6 (2.9)	0.788
Depression (cut off >7), yes (%)	10 (62.5%)	54 (47.4%)	0.295
Depression (cut off >10), yes (%)	3 (18.8%)	20 (17.5%)	1.00
Leg cramp, yes (%)	15 (93.8%)	85 (74.6%)	0.117
Leg cramp, how often, yes (%)			0.038
More than once a day	4 (25.0%)	5 (4.4%)	
Once a day	2 (12.5%)	16 (14.0%)	
Once every several days	3 (18.8%)	32 (28.1%)	
Once every several weeks	4 (25.0%)	21 (18.4%)	
Once every several months	2 (12.5%)	11 (9.6%)	
None	1 (6.3%)	29 (25.9%)	
Leg cramp, when, yes (%)			0.381
During sleep	8 (50.0%)	54 (47.4%)	
In the morning	3 (18.8%)	20 (17.5%)	
In the afternoon	1 (6.3%)	3 (2.6%)	
In the evening	2 (12.5%)	6 (5.3%)	
Non-regular time	1 (6.3%)	2 (1.8%)	
None	1 (6.3%)	29 (25.9%)	
Leg cramp, where, yes (%)			0.042
Calf	10 (62.5%)	55 (48.2%)	
Shin	1 (6.3%)	6 (5.3%)	
Anterior thigh	2 (12.5%)	1 (0.9%)	
Posterior thigh	1 (6.3%)	9 (7.9%)	
Foot	1 (6.3%)	14 (12.3%)	
None	1 (6.3%)	29 (25.9%)	

LSS; lumbar spinal stenosis, RLS; restless legs syndrome, JOA; Japanese Orthopaedic Association, VAS; Visual analogue scale, IRLSSG; International RLS Study Group, HADS-D; Depression subscales of Hospital Anxiety and Depression Scale.

## Data Availability

Data are contained within the article.

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
