# Peer review of "Prevalence of Restless Legs Syndrome and its Symptoms among Patients with Spinal Disorders"

_jcm, 2021, doi:10.3390/jcm10215001_

Round 1
Reviewer 1 Report
The title is prevalence of restless legs syndrome and its symptoms among patients with spinal disorders. Authors should emphasize the discussion of the occurrence of RLS symptoms in spinal disorders patients, as well as possible link between spinal disorders and RLS symptoms.
The critical symptom of RLS is an urge to move the leg (IRLSSG criteria 1). Thus, spinal disorders patients expressing IRLSSG 1, or 1+2, or 1+3, or 1+2+3, can be considered as RLS(-) with RLS symptoms. The total number of spinal disorders without RLS, but with symptoms of IRLSSG criteria 1 shown in table 1 is 124, which is contradicted with the text showing 207 patients met one or more of the RLS criteria.
A comparison of symptoms, such as anxiety, depression, IRLSSG questionnaire score, and so on, between before and after surgery in RLS and in RLS(-) patients should also be done to evaluate whether surgery improves symptoms. It seems that RLS symptoms are not improved after surgery in LSS+RLS patients in the current study. This is different from Kocabicak et al’s study. The authors suggested that RLS could be the residual symptoms after surgery. However, the results do not support the hypothesis.
In discussion, the authors said that patients do not complete the RLS questionnaire before and after surgery. Where is the data shown in table 2 coming from?
Lower limb pain is worsened by walking and is relieved by sitting in LSS patients. In contrast, the uncomfortable sensations in the leg can be relieved by walking and worsened by sitting in RLS patients. How LSS+RLS patients and/or LSS-RLS(-) patients who had at least one criterion of RLS react to lower limb pain and uncomfortable sensations? The painful sensation is one of the factors that cause RLS, but RLS do not induce pain. Furthermore, there is no evidence showing the mesolimbic dopamine system is decreased in RLS patients.
Tables:
The format of the first item, variables, should be changed. It is difficult to read.
The number shown in the table is confusing. Some may represent %, and some may represent average and standard deviation. Please clarify.
The number of female patients should also be present.
Table 1
The number of RLS patients listed in the category, cervical spondylotic myelopathy, is 43. However, the total number of RLS patients is 32. Please explain.
The total number of lumbar spinal disorders patients is 234, with 26 patients having RLS and 208 patients are RLS negative.
Table 2
Table 2 shows the outcome of 120 LSS patients who underwent lumbar decompression surgery. Whereas, the total number of patients underwent surgery shown in table 1 and text is 218 and 130, respectively. Please clarify.
How many LSS-RLS(-) patients underwent surgery have RLS symptoms before and after surgery?
Author Response
Thank you for revising my manuscript and giving valuable suggestions. I hope I could reply all the questions suggested by the reviewer adequately. Please check my revised manuscript and give full-consideration for publication.
The title is prevalence of restless legs syndrome and its symptoms among patients with spinal disorders. Authors should emphasize the discussion of the occurrence of RLS symptoms in spinal disorders patients, as well as possible link between spinal disorders and RLS symptoms.
Response:Thank you for your comment. We deleted some sentences that are not focus on spinal disorder and rewrote the discussion section short.
The critical symptom of RLS is an urge to move the leg (IRLSSG criteria 1). Thus, spinal disorders patients expressing IRLSSG 1, or 1+2, or 1+3, or 1+2+3, can be considered as RLS (-) with RLS symptoms. The total number of spinal disorders without RLS, but with symptoms of IRLSSG criteria 1 shown in table 1 is 124, which is contradicted with the text showing 207 patients met one or more of the RLS criteria.
Response:Thank you for your comment. The meaning of the number 207 indicates the total number of participants who answered yes at least in the IRLSSG questionnaire. (51+70+54+32=207) The subjects who answered no to criteria 1 but no to the other criteria were included in the 207 patients.  
A comparison of symptoms, such as anxiety, depression, IRLSSG questionnaire score, and so on, between before and after surgery in RLS and in RLS (-) patients should also be done to evaluate whether surgery improves symptoms. It seems that RLS symptoms are not improved after surgery in LSS+RLS patients in the current study. This is different from Kocabicak et al’s study. The authors suggested that RLS could be the residual symptoms after surgery. However, the results do not support the hypothesis.
Response:Thank you for your important comments. As you mentioned above, the current research design cannot solve the research question whether RLS symptoms improve with the improvement of spinal disorders. This is one of the limitations of our study and we mentioned it in the discussion section. According to the LSS-part (Table 2), we investigated the prevalence of RLS symptoms at the final follow up. the prevalence of RLS after surgery was 24.2% in the Kocabicak et al reports and 12.3% in our study. These results support our hypothesis that RLS symptoms are relatively common in spine patients and RLS must always be considered in the differential diagnosis of patients who are resistant to conservative treatment or those with residual symptoms in the lower extremities after surgery.
In discussion, the authors said that patients do not complete the RLS questionnaire before and after surgery. Where is the data shown in table 2 coming from?
Response:Thank you for your comments. The data shown in the Table 2 come from the latest follow up not before surgery. We added the information in the results section.
The results of the comparison are shown in Table 2. Of the 120 LSS patients who underwent decompression, 16 (12.3%) had RLS at the final follow up.
Lower limb pain is worsened by walking and is relieved by sitting in LSS patients. In contrast, the uncomfortable sensations in the leg can be relieved by walking and worsened by sitting in RLS patients. How LSS+RLS patients and/or LSS-RLS (-) patients who had at least one criterion of RLS react to lower limb pain and uncomfortable sensations? The painful sensation is one of the factors that cause RLS, but RLS do not induce pain. Furthermore, there is no evidence showing the mesolimbic dopamine system is decreased in RLS patients.
Response:Thank you for your comments. Unfortunately, we don’t have detailed data on how patients react for lower limb pain and uncomfortable sensations. However, we speculate that patients often respond to postoperative residual symptoms by trying to move their legs. More than 50% LSS patients without RLS after surgery had an urge to move their legs in our study. We have deleted the words of mesolimbic from the text about RLS.
We had hypothesized that the low volume of dopamine associated with chronic pain and RLS or RLS-like symptoms may affect the clinical outcomes after surgery.
Tables:
The format of the first item, variables, should be changed. It is difficult to read. The number shown in the table is confusing. Some may represent %, and some may represent average and standard deviation. Please clarify. The number of female patients should also be present.
Response: Thank you for your suggestion. We have added % to the frequency to avoid confusion. I also added the number of female patients.
Table 1
The number of RLS patients listed in the category, cervical spondylotic myelopathy, is 43. However, the total number of RLS patients is 32. Please explain.
Response: Thank you for your suggestion. The numbers to be entered were replaced with and without RSL. I will fix it.
The total number of lumbar spinal disorders patients is 234, with 26 patients having RLS and 208 patients are RLS negative.
Response: Thank you for your suggestion. The number of degenerative lumbar spondylolisthesis (DS) and degenerative lumbar scoliosis (DLS) were included in the number of lumbar spinal stenosis (LSS). Because the term of LSS in table 2 included DS and DLS patients. I fixed the data in Table 1 to avoid confusion.
Table 2
Table 2 shows the outcome of 120 LSS patients who underwent lumbar decompression surgery. Whereas, the total number of patients underwent surgery shown in table 1 and text is 218 and 130, respectively. Please clarify.
Response: Thank you for your suggestion. We analyzed the 130 LSS patients who underwent lumbar decompression surgery. I fixed the date from 120 to 130 in text of the results section. 218 means the number of patients who have had surgery including the all levels, such as lumbar (169), thoracic (3), and cervical spine (46) and all type of disease. 152 patients including LSS, DS and DLS were performed lumber decompression surgery and 22 were excluded for incomplete pre or postoperative data. Finally, we analyzed the 130 patients.
How many LSS-RLS(-) patients underwent surgery have RLS symptoms before and after surgery?
Response: Thank you for your suggestion. Our study was cross sectional study, therefore we don’t have the data about RLS both before and after surgery.

Reviewer 2 Report
Very nice presented study, backround and results. Discussion is too long and should focus more on the findings in the study
- Currently there are 5 RLS diagnostic criteria and all must be met, the fifth criteria = The occurrence of the above features are not solely accounted for as symptoms primary to another medical or a behavioral condition (e.g., myalgia, venous stasis, leg edema, arthritis, leg cramps, positional discomfort, habitual foot tapping), according to Internation RLS study group. It is important to rule out sk RLS "mimics"
- RLS can be triggered as s side effect of drugs and there is lack of informatin about medicines in the article
- There are no information about other comorbid conditions in the subjects such as diabetes or cardiovascular conditions, polyneuropathy.
- Results: Flow chart on the inclusion process and follow-up should be added
- Figure on RLS/no RLS and pain intensity
- The study would increase its significance if RLS were measured both before and after the surgery to examine the development of RLS symptoms and severity of RLS with IRLSSG rating scale
Author Response
Thank you for revising my manuscript and giving valuable suggestions. I hope I could reply all the questions suggested by the reviewer adequately. Please check my revised manuscript and give full-consideration for publication.
Very nice, presented study, background, and results. Discussion is too long and should focus more on the findings in the study
Response:Thank you for your comment. We deleted some sentences that are not focus on spinal disorder and rewrote the discussion section short.
Total words 1213 to 1102
- Currently there are 5 RLS diagnostic criteria and all must be met, the fifth criteria = The occurrence of the above features are not solely accounted for as symptoms primary to another medical or a behavioral condition (e.g., myalgia, venous stasis, leg edema, arthritis, leg cramps, positional discomfort, habitual foot tapping), according to International RLS study group. It is important to rule out sk RLS "mimics"
- RLS can be triggered as s side effect of drugs and there is lack of information about medicines in the article
- There are no information about other comorbid conditions in the subjects such as diabetes or cardiovascular conditions, polyneuropathy.
Response: We agree with your comments. ‘RLS/WED mimics’ have been commonly confused with RLS particularly in surveys because they produce symptoms that meet or at least come very close to meeting criteria 1–4. We are aware that most of the studies are likely to be mimics RLS because all cases are spinal disorders, and we only investigated the prevalence of “RLS and its symptoms” among patients with spinal disorders and the impact of “RLS symptoms” on the clinical outcomes after LSS surgery. We added these limitations in the discussion section.
We performed this survey using the NIH/IRLSSG criteria 1-4. The occurrence of RLS symptoms is not solely accounted for as symptoms primary to another medical, side effect of drugs or a behavioral condition.
- Results: Flow chart on the inclusion process and follow-up should be added
Response: Thank you for your comments. Our study was cross sectional study recruited spine clinic outpatients to screening for RLS, therefore, there is no follow-up. The inclusion criteria are as described in the method. age > 65 years, diagnosis of a spinal disorder, and the ability to ambulate.
- Figure on RLS/no RLS and pain intensity
Response: Thank you for your comments. As shown in Table 2, LSS with RLS patients tended to have more back pain (33.8 vs 19.0; p=0.02), leg pain (24.6 vs 20.7) and leg numbness (39.7 vs 29.2; P=0.21). We have no information about the patients with cervical and thoracic disorders.
- The study would increase its significance if RLS were measured both before and after the surgery to examine the development of RLS symptoms and severity of RLS with IRLSSG rating scale
Response: Thank you for your comment. This point is one of limitations of our research. Our results do not establish a cause-effect relationship between RLS and spinal disorders. We also thought that if these points are resolved, our paper will become a more clinically meaningful.

Reviewer 3 Report
An interesting work. However, I have several suggestion to improve the quality of the manuscript:
1) In page 2 the authors stated that "RLS is a frequent neuropathy". RLS is not a neuropathy, although several causes of neuropathy can be associated with RLS.
2) Page 3, reference [20]. This reference corresponde to a scale of severity of RLS, not to the diagnostic criteria.
3) Two more recent citations regarding association of RLS with peripheral neuropathy (PMID: 33772991; doi: 10.1111/ene.14840) and with Parkinson's disease and other movement disorders (PMID: 31004074; doi: 10.1212/WNL.0000000000007500) could be included among the citations.
4) The prevalence found or RLS in this series of patients with spinal disease is among the ranges of prevalence found in the normal population. This should be emphasized, since control group is lacking.
5) It should be very interesting to describe in how many patients RLS symptoms improved or dissapeared after spinal surgery. This would suggest casuality.
6) It should be useful to describe the response of RLS symptoms to dopamine agonists, gabapentin, clonazepam or other medical therapies.
7) How many patients had positive family history of RLS?
Author Response
Thank you for revising my manuscript and giving valuable suggestions. I hope I could reply all the questions suggested by the reviewer adequately. Please check my revised manuscript and give full-consideration for publication.
An interesting work. However, I have several suggestion to improve the quality of the manuscript:
1) In page 2 the authors stated that "RLS is a frequent neuropathy". RLS is not a neuropathy, although several causes of neuropathy can be associated with RLS.
Response: Thank you for your suggestion. We have changed the sentence.
“RLS is frequently associated with painful polyneuropathy”
2) Page 3, reference [20]. This reference corresponded to a scale of severity of RLS, not to the diagnostic criteria.
Response: Thank you for your suggestion. We deleted ref 20 from the text.
3) Two more recent citations regarding association of RLS with peripheral neuropathy (PMID: 33772991; doi: 10.1111/ene.14840) and with Parkinson's disease and other movement disorders (PMID: 31004074; doi: 10.1212/WNL.0000000000007500) could be included among the citations.
Response: Thank you for your suggestion. We replaced our ref 7 and 8 with your suggesting citation.
Peripheral neuropathy
Jiménez-Jiménez FJ, Alonso-Navarro H, García-Martín E, Agúndez JAG. Association between restless legs syndrome and peripheral neuropathy: A systematic review and meta-analysis. Eur J Neurol2021 Jul;28(7):2423-42.
Parkinson's disease
Alonso-Navarro H, García-Martín E, Agúndez JAG, Jiménez-Jiménez FJ. Association between restless legs syndrome and other movement disorders. Neurology2019 May 14;92(20):948-64.
4) The prevalence found or RLS in this series of patients with spinal disease is among the ranges of prevalence found in the normal population. This should be emphasized, since control group is lacking.
Response: Thank you for your suggestion. We revised the text to reflect your suggestions.
We speculated that the prevalence of RLS in our patients with spinal disorders was among the range of prevalence found in the normal population, while higher than that in the Japanese general population.
5) It should be very interesting to describe in how many patients RLS symptoms improved or dissapeared after spinal surgery. This would suggest casuality.
Response: Thank you for your suggestion. Unfortunately, the current research design is cross sectional study, therefore, cannot solve this research question whether RLS symptoms improve with the improvement of spinal disorders. We also think further studies are required. This is one of the limitations of our study and we mentioned it in the discussion section.
6) It should be useful to describe the response of RLS symptoms to dopamine agonists, gabapentin, clonazepam or other medical therapies.
Response: Thank you for your suggestion. We added the following sentence about the cases where the symptom was improved by dopamine agonists in this series.
In our study, 32 patients met 4 IRLSSG criteria but only six were treated, for example, with dopaminergic agents. Among these six patients, five experienced an improvement of their residual symptoms after surgery on medication. The awareness of RLS can help reduce diagnostic errors and it can be treated quite easily with some medications.
7) How many patients had positive family history of RLS?
Response: Thank you for your suggestion. Unfortunately, we didn’t investigate about the family history of RLS.

Round 2
Reviewer 3 Report
No additional comments